# Identification of Chalcone Isomerase Family Genes and Roles of *CnCHI4* in Flavonoid Metabolism in *Camellia nitidissima*

**DOI:** 10.3390/biom13010041

**Published:** 2022-12-26

**Authors:** Suhang Yu, Jiyuan Li, Ting Peng, Sui Ni, Yi Feng, Qiushi Wang, Minyan Wang, Xian Chu, Zhengqi Fan, Xinlei Li, Hengfu Yin, Wanchuan Ge, Weixin Liu

**Affiliations:** 1Key Laboratory of Tree Breeding of Zhejiang Province, Research Institute of Subtropical Forestry, Chinese Academy of Forestry, Hangzhou 311400, China; 2School of Marine Sciences, Ningbo University, Ningbo 315800, China; 3Jinhua Moxian Horticultural Engineering Co., Ltd., Jinhua 321000, China; 4College of Agriculture, Guizhou University, Guiyang 550525, China; 5Changchun GeneScience Pharmaceuticals Co., Ltd., Changchun 130103, China

**Keywords:** *Camellia nitidissima*, chalcone isomerase, flavonoids, *CnCHI4*, MYB

## Abstract

*Camellia nitidissima* is a woody plant with high ornamental value, and its golden-yellow flowers are rich in a variety of bioactive substances, especially flavonoids, that are beneficial to human health. Chalcone isomerases (CHIs) are key enzymes in the flavonoid biosynthesis pathway; however, there is a scarcity of information regarding the *CHI* family genes of *C. nitidissima*. In this study, seven *CHI* genes of *C. nitidissima* were identified and divided into three subfamilies by phylogenetic analysis. The results of multiple sequence alignment revealed that, unlike CnCHI1/5/6/7, CnCHI2/3/4 are bona fide CHIs that contain all the active site and critical catalytic residues. Analysis of the expression patterns of *CnCHIs* and the total flavonoid content of the flowers at different developmental stages revealed that *CnCHI4* might play an essential role in the flavonoid biosynthesis pathway of *C. nitidissima*. *CnCHI4* overexpression significantly increased flavonoid production in *Nicotiana tabacum* and *C. nitidissima*. The results of the dual-luciferase reporter assay and yeast one-hybrid system revealed that CnMYB7 was the key transcription factor that governed the transcription of *CnCHI4*. The study provides a comprehensive understanding of the *CHI* family genes of *C. nitidissima* and performed a preliminary analysis of their functions and regulatory mechanisms.

## 1. Introduction

Flavonoids are an important class of plant secondary metabolites that contain a characteristic phenylchromenone core [1]. They are widely distributed all over the plant kingdom and play an important role in the formation of plant pigments, allelopathy [2], signaling [3], and fertility [4]. Flavonoids have vital roles in plant development and are beneficial to human health owing to various bioactivities [5], including antioxidant [6], anti-inflammatory, and anti-cancer properties [7].

Flavonoids can be subdivided into different classes according to their chemical structures, including flavanones, flavones, flavonols, proanthocyanidins, and anthocyanins [8]. The flavonoid biosynthesis pathway comprises a cascade of enzymatic reactions; chalcone synthases (CHSs) and chalcone isomerases (CHIs) catalyze the first two committed steps of the pathway and provide essential precursors for downstream flavonoid synthesis [9]. CHSs catalyze the production of 2′,4,4′,6′-tetrahydroxychalcone [10], which is converted to the corresponding (2S)-flavanones following catalysis by CHIs [11]. Although the latter step can occur spontaneously, CHIs can elevate the rate of the reaction by 10^7^-fold [12].

Plant CHIs are categorized into four types, namely, types I, II, III, and IV. Of these, type I and type II CHIs are catalytically active and represent bona fide CHIs, while the catalytic activity is lost in type III and type IV CHIs. Type I CHIs are ubiquitous in vascular plants and catalyze the conversion of 6′-hydroxychalcone into 5-hydroxyflavanone, while type II CHIs are primarily found in leguminous plants and have a broader substrate specificity [1,13]. In addition to catalyzing the conversion of 6′-hydroxychalcone to 5-hydroxyflavanone, type II CHIs catalyze the conversion of 6′-deoxychalcone to 5-deoxyflavanone [14,15]. It has been reported that a lack of CHI activity in onion leads to the inhibition of the flavonoid biosynthesis pathway, which results in the accumulation of chalcone derivatives [16]. On the contrary, the overexpression of *CHI* increases the production of downstream products, including flavonols [17]. Previous studies have demonstrated that type III CHIs function as fatty-acid-binding proteins (FAPs) and are involved in fatty acid metabolism [18]. Type III CHIs are also regarded as progenitors of bona fide CHIs [18]. Type IV CHIs are also known as chalcone isomerase-like (CHIL) proteins that lack catalytic activity owing to several substitutions in the catalytic residues [10]. Few studies have investigated the functions of type IV CHIs, and recent studies have demonstrated that CHIL proteins can increase the yield of flavonoids by interacting with CHIs and CHSs [19,20,21].

Flavonoid biosynthesis in plants is tightly regulated by several transcription factors, especially the MYB transcription factors. It has been reported that the MdMYB308L transcription factor can positively regulate the accumulation of anthocyanins in apple [22]. The PavMYB10.1 transcription factor binds to the promoter regions of the anthocyanin biosynthesis genes, *PavANS* and *PavUFGT*, in sweet cherry [23]. Another study identified AtMYB12 as a flavonol-specific activator of flavonoid biosynthesis in *Arabidopsis thaliana*. [24]. MYB transcription factors bind to bHLH and WD40 transcription factors to form the MBW complex that regulates flavonoid biosynthesis in several plant species [25]. An increasing number of transcription factors that regulate flavonoid biosynthesis have been identified to date [26].

*C. nitidissima* is also known as ‘Giant Panda of the Plant Kingdom’ and ‘the Queen of Camellia’ owing to its golden-yellow flowers [27]. *C. nitidissima* contains a variety of biologically active compounds, especially flavonoids, which are beneficial to human health and have been studied extensively [28,29]. Some results suggested that several structural genes play critical roles in the formation of flavonoids in *C. nitidissima* [30,31], and it was found that two *CHI* genes were involved in flavonoid biosynthesis during the early stages of flower development. *CHI* genes have been extensively studied in several species of plants, especially *A. thaliana* and *Glycine max*; however, there is little information regarding the *CHI* genes of *C. nitidissima*. In this study, a total of 7 *CHI* genes of *C. nitidissima* were identified and classified, and their conserved domains and expression profiles were subsequently determined. The findings revealed that *CnCHI4* may play pivotal roles in flavonoid metabolism in *C. nitidissima*, and the transcriptional regulation of *CnCHI4* was additionally investigated.

## 2. Materials and Methods

### 2.1. Identification of CHI Family Genes in C. nitidissima

The transcriptome sequence data used in this study were previously generated and validated in our laboratory (PRJNA909942) [32]. The transcriptome data of *C. nitidissima* were simultaneously obtained from seven tissue types, including roots, leaves, fruits, sepals, petals, stamens, and pistils. Also, the RNA-seq data were obtained from flower samples at different stages of development, including buds that were 10 mm, 20 mm, and 30 mm in diameter, denoted as stages B1, B2, and B3, respectively, and half-open (Fh) and completely open (Fc) flowers. Total RNAs were extracted by RNA Extraction Kit DP441 (TIANGEN, Beijing, China), and Agilent 2100 (Agilent Technologies, Santa Clara, CA, USA) was used to analyze the quality of RNA. mRNA was collected using oligo-attached poly-T magnetic beads. Random hexamer primers were used for cDNA synthesis; the sequencing library was created via polymerase chain reaction (PCR) amplification. Then, cDNA libraries were sequenced on Illumina HiSeq Xten (Illumina, San Diego, CA, USA), and clean data were acquired by removing adapter reads, and low-quality reads from raw data. Additionally, mRNA was reverse transcribed into full-length cDNA using a SMARTer™ PCR cDNA Synthesis Kit (Clontech, Mountain View, CA, USA), and PacBio RS II (Pacific Biosciences, Menlo Park, CA, USA) was used for RNA sequencing. ToFu pipeline with default parameters was used to convert raw reads into error-corrected reads of insert (ROIs). By looking for the polyA tail signal and the 5′ and 3′ cDNA primers in ROIs, full-length non-chemiric (FLNC) transcripts were identified. To obtain consensus isoforms, ICE (Iterative Clustering for Error Correction) was conducted, and the results were polished by Quiver. TransDecoder v3.0.0 was used to predict the ORFs (open reading frames). Bowtie2 was used to align sequencing reads from second-generation sequencing against the reads of third-generation sequencing, and transcript abundance quantification was obtained using RSEM and presented as Fragments Per Kilobase of transcript per Million mapped reads (FPKM). The formula is shown as follows: cDNA Fragments/(Mapped Fragments (Millions) × Transcript Length (kb)).

Hidden Markov models of the CHIs were identified from the Pfam database (http://pfam-legacy.xfam.org/, accessed on 17 November 2021) (accession IDs: PF02431, PF16035, and PF16036), and the models were used as query against the transcriptome database for identifying candidates belonging to the *CHI* gene family with an E-value of 1 × 10^−5^, using the TBtools program [33]. The sequences of the proteins encoded by the *CHI* family genes of *A. thaliana* (*AtCHI*, *AtCHIL*, *AtFAP1*, *AtFAP2*, and *AtFAP3*) were then subjected to BLASTp search using a threshold E-value of < 1 × 10^−20^.

The presence of the true CHI domain in these proteins was determined using the Conserved Domain (https://www.ncbi.nlm.nih.gov/cdd, accessed on 29 November 2021) and SMART (http://smart.embl-heidelberg.de/, accessed on 29 November 2021) databases.

### 2.2. Structural and Biochemical Analyses of the CHI Proteins of C. nitidissima

The number of amino acids, molecular weight, and theoretical isoelectric point (pI) of the CnCHI proteins were determined using the Expasy webserver (https://web.expasy.org/protparam/, accessed on 7 December 2021). The SignaIP-5.0 server was used for predicting the signal peptides (https://services.healthtech.dtu.dk/service.php?SignalP-5.0, accessed on 27 December 2021). The secondary structure composition of CnCHI proteins was predicted using SOPMA (https://npsa-prabi.ibcp.fr/cgi-bin/npsa_automat.pl?page=/NPSA/npsa_sopma.html, accessed on 27 December 2021). The conserved motifs were determined with MEME (https://meme-suite.org/meme/tools/meme, accessed on 27 December 2021), and a maximum of 10 motifs were predicted for each protein.

### 2.3. Phylogenetic Analysis and Multiple Sequence Alignment of CnCHI Proteins

The sequences of the CnCHI proteins of *C. nitidissima* were compared with the CHI protein sequences of other plants. The sequences of CHI proteins were retrieved from the NCBI (https://www.ncbi.nlm.nih.gov/, accessed on 10 January 2022). A multiple sequence alignment of the protein sequences was constructed using the MUSCLE program in MEGA11 software with default parameters, and a phylogenetic tree was constructed by the neighbor-joining method with 1000 bootstrap replications. The phylogenetic tree was visualized with iTOL (https://itol.embl.de/, accessed on 17 April 2022). The multiple sequence alignment was edited using GeneDoc, and Adobe Illustrator software was used for image processing.

### 2.4. Plant Materials and Sample Preparation

The flower and leaf samples of *C. nitidissima* used were obtained from the Germplasm Resource Center of the Institute of Subtropical Forestry, Chinese Academy of Forestry (Daqiao Road, Hangzhou, China), grown in the field and are about 10 years old. For RNA extraction and determination of total flavonoid content (TFC), flower samples at different development stages (B1, B2, B3, Fh, and Fc) were collected and immediately snap-frozen in liquid nitrogen and kept frozen at −80 °C for further use. For transient overexpression, the shoots with leaves about 20 cm were collected and cultivated with clean water.

For DNA extraction, the young leaves of *C. nitidissima* were obtained from the National Camellia Germplasm Resource Bank (Gecun Road, Nanning City, China). Leaf samples were collected and immediately snap-frozen in liquid nitrogen and kept on dry ice, back to the laboratory, and kept frozen at −80 °C.

Tobacco (*N. tabacum*) used in this experiment was grown at 25 °C with a 16-h/8-h (light/dark) cycle.

### 2.5. Determination of Total Flavonoid Content (TFC)

The TFC in fresh samples was quantified by UV-vis spectrophotometry. Tissues of *C. nitidissima* and tobacco were ground in liquid nitrogen and digested in 60% ethanol at 60 °C for 4 h, and TFC detection kit BC1330 (Solarbio, Beijing, China) was used. 

### 2.6. Genes Isolation

RNA extraction was performed using RNA extraction reagent kits DP441 (TIANGEN, Beijing, China), and the cDNA library of *C. nitidissima* was built using a PrimeScript RT Master MIX (Takara, Dalian, China). According to transcriptome sequencing data, primer premier 5 was used for primer design, and the full-length coding sequences (CDS) of genes were amplified from the cDNA library and cloned into pMD19-T vectors (Takara, Dalian, China). The recombinant vector was transferred into DH5α competent cells of *Escherichia coli* (Takara, Dalian, China), and positive clones were confirmed by sequencing. Primers for gene cloning are listed in Appendix A.

### 2.7. Analysis of Subcellular Localization

The pCAMBIA1302 vector with a GFP-tag was used for analyzing the subcellular localization, and the empty vector and vector with the sequences of *CnCHI4* and *CnMYB7* were transformed into the GV3101 strain of *Agrobacterium tumefaciens* (Weidi Bio, Shanghai, China). The positive clones were selected and grown on LB liquid medium to an OD_600_ of 0.6–0.8 and subsequently resuspended in activation buffer (10 mM MgCl_2_, 10 mM MES, and 100 mM acetosyringone; pH 5.8) to an OD_600_ of 0.5. The solutions were injected into the abaxial side of *Nicotiana benthamiana* leaves with a needleless syringe. The tobaccos were cultured in darkness for 24 h (25 °C) and followed with 16 h-light/8 h-dark (25 °C) conditions for 48 h. The fluorescence was measured with a confocal microscope (Carl Zeiss, Oberkochen, Germany). Primer sequences used in vector construction are provided in Appendix A.

### 2.8. CnCHI4 Overexpression in N. tabacum

The full-length CDS of *CnCHI4* was cloned into the pCAMBIA1302 vector, and tobacco plants were transformed with the GV3101 strain (Weidi Bio, Shanghai, China) using the leaf disc method [34]. The transformed tobacco plants were selected using 20 mg/L hygromycin, and transformation was confirmed by polymerase chain reaction (PCR) of the genomic DNA using primers specific for the hygromycin resistance gene (Appendix A). The levels of *CnCHI4* transcript in the leaves of transgenic lines were measured by quantitative real-time PCR (qRT-PCR), the leaves were collected and snap-frozen in liquid nitrogen for analysis of TFC.

### 2.9. Transient Overexpression of CnCHI4 in C. nitidissima

The leaves of *C. nitidissima* were used for transient overexpression. The GV3101 strain of *A. tumefaciens* (Weidi Bio, Shanghai, China) carrying the pCAMBIA1302 or pCAMBIA1302-*CnCHI4* vectors was injected into the left and right blades of each leaf with a needleless syringe, the specific method is the same as that of subcellular localization. Then, the shoots were kept in the dark for 24 h (25 °C) and maintained under 16 h light/8 h-dark (25 °C) conditions for 96 h. The leaves were collected after 5 days of injection and snap-frozen in liquid nitrogen for qRT-PCR and analysis of TFC.

### 2.10. qRT-PCR Analysis

The total RNA from the leaves of *C. nitidissima* and *N. tabacum* was extracted using RNA extraction reagent kits DP441 (TIANGEN, Beijing, China). The total RNA was reverse transcribed to cDNA using a PrimeScript RT Master Mix (Takara, Dalian, China). qRT-PCR analyses were conducted as previously described [35], and the 2^−ΔΔCT^ method was used for quantitative analysis. The sequences of the primers used for qRT-PCR are depicted in Appendix A.

### 2.11. Weighted Gene Co-Expression Network Analysis (WGCNA) and Identification of Differentially Expressed Genes (DEGs)

WGCNA analysis was performed with the WGCNAShiny plugin in TBtools [33], and a total of 7993 genes were selected for subsequent analysis after data filtering (sample percentage = 0.9, expression cutoff = 1, filter method = MAD, and number of reserved genes = 8000). A soft-thresholding power of 16 was used to obtain a scale-free topology index greater than 0.8. The minimum module size was set to 30, and the modules were merged at a cut height of 0.25. The FPKM values of *CnCHI4* at different developmental stages were used for calculating the Pearson correlation with the modules. The modules with a high correlation coefficient were selected for subsequent analysis, and genes with a Module Membership (MM) ≥ 0.75 and Gene Significance (GS) ≥ 0.75 were identified as hub genes. DEGs between stages B2 and B3 were determined with DESeq, and the genes were considered to be differentially expressed if Fold Change ≥ 2 and FDR (False Discovery Rate) < 0.01. The transcription factors were predicted using the PlantTFDB database (http://planttfdb.gao-lab.org/index.php, accessed on 10 April 2022), and the data were visualized using the ggplot2, PerformanceAnalytics, and ggvenn packages in R studio.

### 2.12. Analyses of Promoter Clones and Cis-Elements

The genomic DNA was extracted using the modified CTAB method. Nested PCR was performed using primers specific for the CDS sequences of *CnCHI4* and the promoter sequences of *Camellia japonica CHI*. The primers used for nested PCR are enlisted in Appendix A. The PCR products were ligated into a pMD19-T vector (Takara, Dalian, China), and the resulting plasmids were subsequently transformed into *DH5α* competent cells of *E. coli* (Takara, Dalian, China). The positive clones were selected and analyzed by DNA sequencing. The PlantCARE website was used for cis-elements analysis (http://bioinformatics.psb.ugent.be/webtools/plantcare/html, accessed on 7 March 2022).

### 2.13. Dual-Luciferase Assay

The full-length CDS of *CnMYB7* was cloned into a pGreenII62-SK plasmid, and the promoter sequence of *CnCHI4* was incorporated into a pGreenII0800-LUC plasmid; the primer sequences used to construct vectors are shown in Appendix A. The empty pGreenII62-SK vector was used as the control. The recombinant vectors were transferred into the GV3101 strain of *A*. *tumefaciens* (Weidi Bio, Shanghai, China) and prepared using the same method used for analyzing subcellular localization. The GV3101 cell suspensions containing the pGreenII62-SK and pGreenII0800-LUC vectors were mixed at a ratio of 1:1 and injected into the epidermis of tobacco leaves with a needleless syringe. The enzymatic activities of firefly luciferase (LUC) and renilla luciferase (REN) were determined using luciferase assay reagent kits (Beyotime, Shanghai, China) after three days of injection.

### 2.14. Yeast One-Hybrid Assay

The *CnMYB7* gene was integrated into the pGADT7 plasmid, and the promoter sequences of *CnCHI4* were incorporated into the pHIS2 plasmid; primer sequences are listed in Appendix A. The pGADT7-*Gus* vector was used as the control. The pHIS2 and pGADT7 vectors were co-transformed into the *Saccharomyces cerevisiae* Y187 strain. The cells were grown on SD/-Trp/-Leu/-His medium for 3 days at 30 °C, positive colonies were transferred to the SD/-Trp/-Leu/-His plates supplemented with 275 mg/L 3-Amino-1,2,4-triazole (3-AT).

### 2.15. Statistical Analyses

The statistical analyses were performed with GraphPad Prism 9 (Graphpad, San Diego, CA, USA). All the experiments were performed in triplicate, and the error bars represented the standard error (SE). Statistically significant differences were calculated using Student’s *t*-test at a confidence level of 95.0% (* *p* < 0.05) and 99.0% (** *p* < 0.01).

## 3. Results

### 3.1. Identification and Characterization of CnCHI Proteins

A total of 14 putative proteins were identified using the Hidden Markov models of CHIs (accession IDs: PF02431, PF16035, and PF16036) as a query against the transcriptome database of *C. nitidissima*. The putative proteins were confirmed by conserved domain search against the SMART and NCBI conserved domain databases, and the sequences that contained the complete conserved domain of CHIs were selected. The redundant sequences were finally removed, and seven proteins were classified as members of the CHI family (Table 1).

The number of amino acids, pI, molecular weight, hydropathicity, and instability index were calculated using the Expasy web server, and the results are provided in Table 1. The CnCHI proteins comprised 209–452 amino acids. The molecular weight of these proteins ranged from 23,193.41 to 50,117.33 KDa, of which the molecular weight of CnCHI1 was the lowest owing to relatively fewer amino acid residues, while the molecular weight of CnCHI7 was the highest. The pI of the predicted protein sequences ranged from 4.83 to 8.73. The hydropathicity index of all the CnCHI proteins was less than 0, indicating that all the proteins were hydrophobic. The instability index of all the proteins was higher than 40, indicating that the majority of CnCHI proteins were unstable.

The signal peptides of the seven CnCHI proteins were also predicted using the SignaIP-5.0 server, and the findings revealed that the proteins lacked an N-terminal signal peptide. The secondary structure composition of the CnCHI proteins was predicted with SOPMA (Figure 1A), and the results demonstrated that the proteins secondary structures contained alpha helices, beta turns, random coils, and extended strands.

In order to obtain a better insight into the conservation and diversification of CnCHI proteins, the deduced sequences were submitted to the MEME web server for determining the motif composition. A total of ten motifs, denoted as motifs 1–10, were predicted for each protein, and they are depicted in Figure 1B. The findings revealed that the number of motifs ranged from 2 to 7, and the motifs comprised 15–50 amino acid residues.

### 3.2. Phylogenetic Relationships, Multiple Sequence Alignment, and Classification of CnCHIs

As aforementioned, the CHI proteins of plants can be categorized into four types. A phylogenetic tree was constructed using the neighbor-joining method for evaluating the phylogenetic relationships among the CHI proteins of different species and classifying the CnCHI proteins. As depicted in Figure 2, the proteins were clustered into four clades in the phylogenetic tree, which was consistent with the results of previous studies. The seven CnCHI proteins identified in this study were classified into three types, namely, type I, type III, and type IV proteins. Of these, CnCHI2, CnCHI3, and CnCHI4 were categorized as type I, while CnCHI5, CnCHI6, and CnCHI7 were categorized as type III, and only CnCHI1 was classified as a type IV protein.

The results of multiple sequence alignment (Figure 3) revealed that type I CnCHIs contained all the active site and critical catalytic residues, and the residues were well-conserved in type I and type II CHIs. These findings were consistent with previous studies, which reported that Ser/Ile in type I proteins and Thr/Met in type II proteins determine the differences in substrate preference between these classes [36]. The findings of this study revealed that three homologous type III proteins (CnCHI5, CnCHI6, and CnCHI7) and a type IV protein (CnCHI1) carried the substitutions at nearly all the critical catalytic sites.

### 3.3. Analysis of CnCHI Expression and Measurement of TFC

Analysis of gene expression levels can provide important information for elucidating gene functions. In order to decipher the function of *CnCHI* genes during flower development, the expression levels of seven *CnCHI* genes at different stages of flower development (B1, B2, B3, Fh, and Fc) are displayed in a heatmap. As depicted in Figure 4A, the expression levels of type I *CnCHI* genes varied markedly across the different stages of flower development. The expression levels of *CnCHI2* and *CnCHI3* were very low and negligible compared to that of *CnCHI4*, which indicated that *CnCHI2/3* do not play an important role during the different stages of flower development. The expression level of *CnCHI4* rapidly peaked at the B2 stage. Interestingly, the expression levels of *CnCHI4* reduced markedly during the transition from B2 to B3 stages.

The expression of type III *CnCHI* genes altered during the different stages of flower development. The expression of *CnCHI5* altered in a bimodal manner over time in that the expression of *CnCHI5* was upregulated during the transition from B1 to B2, reduced sharply during the transition from B2 to B3, upregulated from B3 to Fh and declined to nearly 0 during the transition from Fh to Fc. The expression level of *CnCHI6* increased gradually from B1 to Fh, and declined smoothly at the end of flower development. Interestingly, the changes in the expression levels of *CnCHI7* were the complete reverse of those of *CnCHI6*.Analysis of the expression of type IV *CnCHIs* revealed that the expression of *CnCHI1* decreased substantially from B1 to B3 and was maintained at relatively low levels in the subsequent stages. 

Analysis of the expression of *CnCHI* genes in the different tissues, including leaves, roots, fruits, petals, sepals, stamens, and pistils, revealed that the patterns of gene expression were different among the different tissues (Figure 4B). It was evident that the expression of *CnCHI3* was very low in all the tissues, while the expression of *CnCHI4* was generally high across all tissue types. The expression of *CnCHI4* was higher in the leaves, followed by fruits and roots, and was lowest in the sepals and stamens. The expression of *CnCHI2* was higher in the roots compared to that in the other tissues. Analysis of the expression levels of type III *CnCHIs* revealed a high similarity between the expression of *CnCHI5* and *CnCHI6*, which were relatively high in the sepals, stamens, and petals, and low in the roots and fruits. The type IV *CHI* gene, *CnCHI1*, was abundantly and variably expressed across the different tissues. The expression levels of *CnCHI1* were highest in the fruits and lowest in the stamens. *CnCHI1* expression in the fruits was nearly 23-fold that in the stamens. The expression levels of genes were highly associated with gene function, and *CnCHI1* and *CnCHI4* possibly play a major role in the flavonoid biosynthesis pathway of *C. nitidissima*.

Flavonoids can form complexes with aluminum ions in alkaline nitrite solution, which form characteristic absorption peaks at 470 nm. The TFC in *C. nitidissima* petal was therefore measured using a colorimetric method (Figure 4C,D). The TFC levels increased from B1 to B3, peaked at B3, and declined from B3 to Fc. The TFC was highest at B3, reaching a value of 27.02 mg/g. TFC meet a rapid growth during the transition from B2 to B3, which could correspond to a crucial period of flavonoid biosynthesis in the petals of *C. nitidissima*. Moreover, the TFC at the different stages of flower development appeared to reflect the expression of *CnCHI4*, with some lag.

### 3.4. CnCHI4 Is Localized in the Cytoplasm and Nucleus

Analysis of the subcellular localization of proteins provides vital information for understanding protein function. The subcellular localization of CnCHI4 was therefore investigated in this study (Figure 5). The findings demonstrated that CnCHI4 is located in the cytoplasm and nucleus.

### 3.5. Overexpression of CnCHI4 in C. nitidissma and N. tabacum

In order to further elucidate the biological functions of CnCHI4, *CnCHI4* was transiently overexpressed in the leaves of *C. nitidissma*. The relative expression levels of *CnCHI4* in the leaves following infiltration were analyzed by qRT-PCR. As depicted in Figure 6A, the relative expression levels of *CnCHI4* increased seven-fold compared to that of the respective control groups. Correspondingly, the TFC in fresh leaves was also increased (Figure 6B).

*CnCHI4* was also overexpressed in tobacco plants to verify its role in flavonoid metabolism (Figure 6C). The findings revealed that the TFC of the three independent transgenic lines overexpressing *CnCHI4* was significantly higher than that of the wild-type plants (Figure 6D).

### 3.6. Identification of Key Regulatory Factors of CnCHI4

As depicted in Figure 4B, the expression levels of *CnCHI4* declined immediately during the transition from B2 to B3 stage, which indicated that *CnCHI4* is possibly more tightly regulated in this phase. A total of 199 transcription factors were found differentially expressed between B2 and B3 (Figure 7A) and could play an essential role in the regulation of *CnCHI4* expression.

The potential interactions of the genes were determined by WGCNA, and 7993 genes were selected for hierarchical clustering following data filtering. A total of 25 gene modules were identified and indicated by different colors based on the similarity in gene expression patterns(Figure 7B). The modules were subsequently correlated to the trait phenotype (expression level of *CnCHI4* at different stages of flower development). Calculation of the correlation and significance between the characteristics of the modules and *CnCHI4* expression revealed that the brown module was highly correlated to *CnCHI4* expression (r = 0.94, *p* = 2.1 × 10^−7^) (Figure 7C). The genes in the brown module were therefore subjected to subsequent analyses in this study. The Gene Significance (GS) was determined for measuring the correlation between trait phenotype and the genes in the brown module, and the importance of the genes within the module was evaluated based on the Module Membership (MM) (Figure 7D).

By setting the thresholds for GS and MM to 0.75, a total of 26 pivotal transcription factors were identified in the brown module for subsequent analysis. The common transcription factors determined by differential expression analysis and WGCNA were identified by plotting the results with a Venn diagram (Figure 7E). The Venn diagram revealed that five transcription factors identified by differential expression analysis were encoded by the genes in the brown module and were subjected to further analyses. The Pearson correlation coefficients between the expression levels of *CnCHI4* and the five transcription factors at different stages of flower development were calculated. As depicted in Figure 7F, all the transcription factors were significantly associated, either positively or negativelywith the expression of *CnCHI4*, with correlation coefficients ranging from 0.79 to 0.93. Notably, an MYB transcription was identified, and the PlantTFDB database suggested that this MYB transcription factor shared a high similarity to AtMYB7.

### 3.7. CnMYB7 was Localized in the Nucleus

In this study, CnMYB7 was identified as a key factor in the transcriptional regulation of *CnCHI4*. The subcellular localization of CnMYB7 was analyzed to obtain insights into the molecular function of *CnMYB7*. The results demonstrated that the GFP fluorescence of the empty vector was distributed throughout the cells of tobacco leaves, while the fluorescence due to CnMYB7-GFP was only observed in the nucleus (Figure 8A). The findings demonstrated that CnMYB7 is localized in the nucleus and is likely to play a role in regulating gene transcription.

### 3.8. CnMYB7 Bound and Inhibited the CnCHI4 Promoter

The promoter sequence of *CnCHI4* was cloned to elucidate whether *CnCHI4* is transcriptionally regulated by CnMYB7, and the findings revealed the presence of MYB binding sites in the promoter sequence of *CnCHI4* (Appendix A). A dual-luciferase reporter assay was subsequently performed in which *CnMYB7* was incorporated into a pGreenII62-SK vector, and the promoter sequence of *CnCHI4* was integrated into a pGreenII0800-LUC vector. The findings demonstrated that CnMYB7 significantly repressed the expression of the *CnCHI4* promoter by approximately three-fold compared to that of the control (Figure 8B).

The mechanism underlying the effect of CnMYB7 on the transcription of *CnCHI4* was determined by yeast one-hybrid assays. The findings revealed that CnMYB7 could directly bind to the promoter of *CnCHI4*, indicating that CnMYB7 suppresses *CnCHI4* expression by binding to the promoter region of *CnCHI4* (Figure 8C).

## 4. Discussion

Flavonoids are the main active chemical constituents of *C. nitidissima* and are beneficial to human health. However, flavonoid metabolism in *C. nitidissima* remains poorly understood to date. *CHI* is a key enzyme in the flavonoid biosynthesis pathway. In order to obtain further insights into the mechanism of flavonoid biosynthesis in *C. nitidissima* flowers, the *CHI* family genes of *C. nitidissima* were identified. Previous transcriptomic studies did not reveal major specific roles for *CHI* genes in the flavonoid metabolism of *C. nitidissima* [30,31], although recently, it was shown that these genes are upregulated in the petals during the early stages of flower development [30]. A total of seven genes belonging to the *CHI* gene family were identified in this study. *CHI* family genes have been intensively studied in several plants, including *A. thaliana* and *G. max*. The *CHI* genes of *A. thaliana* can be categorized into four types, of which *AtCHI* is a type I *CHI* gene, *AtFAP1/2/3* are type III *CHI* genes, and *AtCHIL* is a type IV gene [18]. The 12 *CHI* genes of *G. max* are categorized into four types, ranging from types I to IV [37]. Only two types of *CHI* genes, types III and IV, have been identified in *Physcomitrella patens* [1]. Analysis of the phylogenetic tree (Figure 2) revealed that the *CnCHIs* were divided into three types, namely, type I (*CnCHI2/3/4*), type III (*CnCHI5/6/7*), and type IV (*CnCHI1*) genes. Type II *CHI* genes are relatively common in leguminous plants [38], which is absent in *C. nitidissima*. Interestingly, the *CHI* genes of *C. nitidissima* exhibited high similarity to those of *Antirrhinum majus* and always shared the same branch in the phylogenetic tree.

Previous studies have demonstrated that type I and type II CHIs are bona fide enzymes that contain several of the critical catalytic residues [39]. The results of multiple sequence alignment demonstrated that CnCHI2/3/4 possess all the critical catalytic residues, including Ser and Ile at positions 190 and 191, respectively (Figure 3). The finding illustrates that CnCHI2/3/4 are type I CHIs that are indispensable to the flavonoid biosynthetic pathway. Nearly all the critical catalytic sites of type III (CnCHI5/6/7) and type IV (CnCHI1) CHIs were substituted, which suggested the loss of catalytic activity. Previous studies have demonstrated that type III CHIs of *A. thaliana* are FAPs that are associated with fatty acid metabolism [18]. CnCHI5 clustered with AtFAP1, and CnCHI6/7 shared the same clade as AtFAP2, which suggested that CnCHI5/6/7 are likely to function as FAPs [1,18]. Type IV CHIs are also known as CHIL proteins, and recent studies have demonstrated that CHIL proteins can bind to CHSs and function as rectifiers that affect flavonoid metabolism [10,19,40]. The type IV CHI identified in this study (CnCHI1) could have a similar function in *C. nitidissima*; however, further studies are necessary to verify the conjecture.

It has been reported that the expression patterns of *CHI* genes differ across plant tissues owing to distinct functions in the different tissues. The *CHI* genes of *Scutellaria lateriflora* are mostly expressed in the roots [41], which is similar to the expression patterns observed in leguminous plants and could be attributed to the interactions between signal molecules and rhizobia [11,13,42]. The expression levels of the *CHI* genes of *Fagopyrum dibotrys* are high in petals and low in stems [43]. In this study, the *CnCHI1* gene of *C. nitidissima* was mainly expressed in the roots, fruits, and leaves, while the expression of *CnCHI4* was relatively low in the sepals and stamens (Figure 4B). Previous studies on other plants have demonstrated that CHIs partake in the formation of pericarp color and influence the bulb color in onion [44] and seed coat color in *A. thaliana*. [13]. The high content of flavonoids in the leaves of *C. nitidissima* could be correlated to the high expression levels of *CnCHI1/4* [27].

Notably, the transcription of *CnCHI4* (type I *CnCHI*) altered drastically during the transition from B1 to B3, and the expression of *CnCHI4* was extremely high during the transition from B1 to B2 but decreased sharply from B2 to B3 (Figure 4A). The same has been reported in *A*. *majus* [45], citrus plants [46] and in a recent study involved in *C. nitidissima* [30], in which the *CHI* genes are expressed slightly earlier during floral development. In this study, the TFC increased acutely from B2 to B3 and began decreasing after B3 during flower development. The findings revealed that the changes in the expression of TFC lag behind the changes in the expression of *CnCHI4* and could be attributed to the various processes between gene expression and metabolite synthesis, including protein translation and transport [37,47]. Additionally, *CHIs* act as early biosynthetic genes in the flavonoid biosynthesis pathway and provide precursors for the biosynthesis of downstream flavonoids. These findings explain the expression pattern of *CnCHI4* observed in this study. The overexpression of *CnCHI4* increased the TFC in *C. nitidissima* and tobacco. The findings indicated that *CnCHI4* plays a prominent role in flavonoid biosynthesis in *C. nitidissima*.

Transcription factors, especially MYB transcription factors, are known to play essential roles in regulating the flavonoid biosynthesis pathway. For instance, flavonol biosynthesis in *A. thaliana* is regulated by MYB11, MYB12, and MYB111 [48,49]. The MYB90, MYB113, and MYB114 transcription factors are related to the regulation of anthocyanin biosynthesis [50,51]. MYB6, an R2R3 MYB transcription factor, promotes anthocyanin and proanthocyanidin biosynthesis in *Populus tomentosa* [52]. Previous studies have reported that MYB10 [53] and MYB308L [22] induce anthocyanin accumulation in apple. *CHI* genes are also regulated by several MYB transcription factors. The ectopic expression of *EsAN2* in tobacco upregulates the expression of *CHS*, *CHI*, and *ANS* and significantly enhances anthocyanin biosynthesis [54]. The MYB11, MYB12, and MYB111 transcription factors in *A. thaliana* play a vital role in promoting the expression of *CHI* genes [55,56,57]. However, there is a scarcity of information regarding the mechanism of regulation of *CHI* expression in *C. nitidissima*. In this study, WGCNA and differential expression analysis revealed that several transcription factors act as pivotal regulators of *CnCHI4* expression. Using dual-luciferase reporter and yeast one-hybrid assays, the study further verified that the CnMYB7 transcription factor is a major regulator of *CnCHI4* expression and negatively regulates the expression of *CnCHI4*.

## Figures and Tables

**Figure 1 biomolecules-13-00041-f001:**
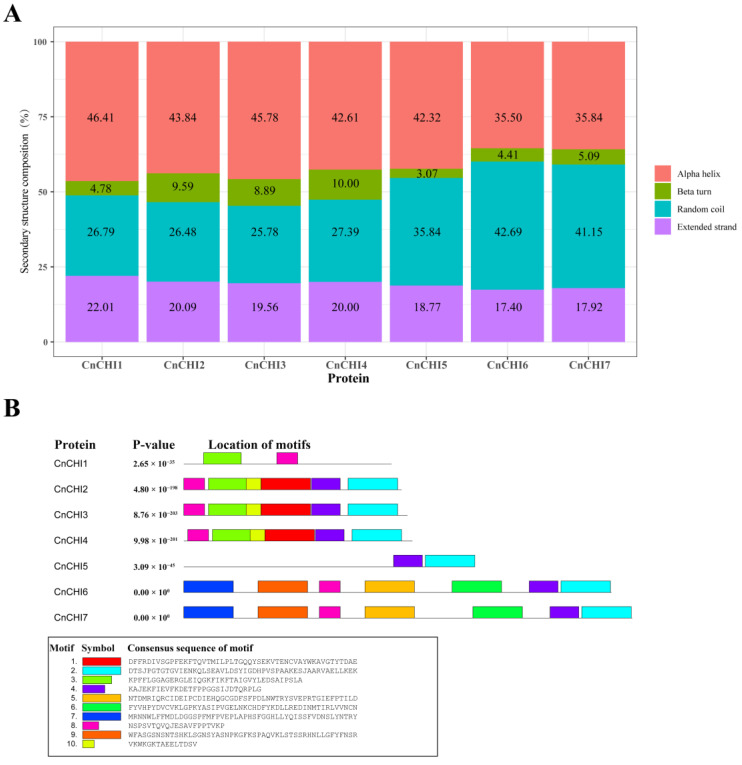
Characteristics of CnCHI protein sequences. (**A**) Secondary structure composition of CnCHI proteins. (**B**) Distribution of the conserved motifs in CnCHI proteins. A total of 10 motifs were predicted for each protein using the MEME online analysis software.

**Figure 2 biomolecules-13-00041-f002:**
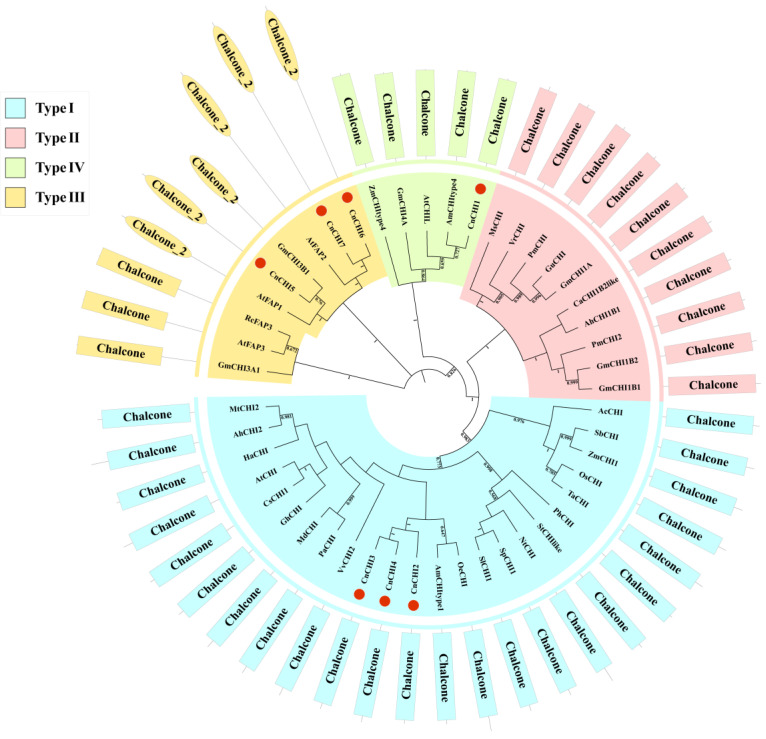
Phylogenetic analyses of the deduced protein sequences of CHIs. The different colors represent different subgroups. Information pertaining to the CHI protein domains was obtained from Pfam and provided in the outer circle of the phylogenetic tree. The GenBank accession numbers are as follows: AtCHI NP_191072.1, NtCHI NP_001312216.1, AtCHIL AAL36093.1, GmCHI1A NP_001235219.1, GmCHI1B1 NP_001236755.1, GmCHI3A1 XP_006593488.1, SlCHI1 NP_001307640.1, VvCHI NP_001268033.1, ZmCHI1 NP_001144002.2, MtCHI2 XP_003592768.2, CsCHI1 XP_010516069.1, SpCHI1 NP_001361335.1, GmCHI4A NP_001236782.1, GmCHI1B2 NP_001236097.2, HaCHI XP_022011225.1, AhCHI1B1 XP_025620961.1, VrCHI NP_001304223.1, TaCHI XP_044391246.1, StCHIlike XP_006348610.1, MdCHI XP_028956659.1, PaCHI XP_021813821.1, AhCHI2 XP_025674780.1, RcFAP3 XP_024199267.1, GmCHI3B1 XP_040869742.1, AtFAP1 NP_567140.1, AtFAP2 Q84RK2.2, AtFAP3 NP_175757.1, GhCHI NP_001314370.2, AmCHItype1 BAO32070.1, OeCHI AHI86006.1, PhCHI CAA32730.1, AcCHI AAS48418.1, OsCHI AAO65886.1, SbCHI XP_002463631, CaCHI1B2like XP_004497327.1, PmCHI BAA09795.1, MsCHI AAB41524.1, PmCHI2 ADV71377.1, GuCHI ABM66533.1, ZmCHItype4 NP_001151452.1, and AmCHItype4 BAO32071.1.

**Figure 3 biomolecules-13-00041-f003:**
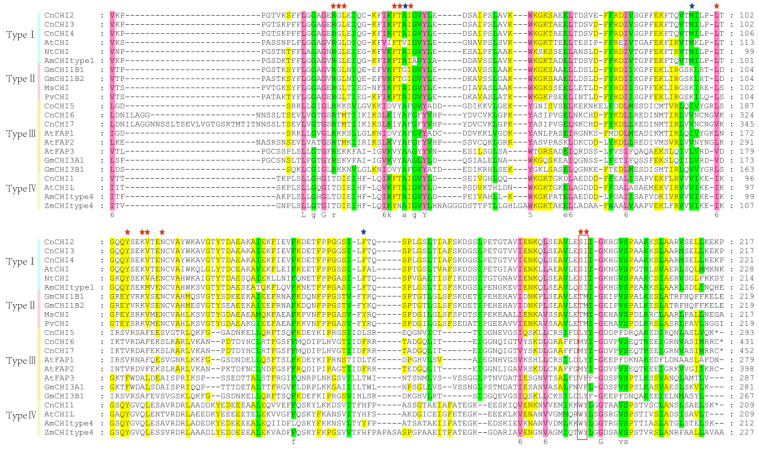
Multiple sequence alignment of the CHI proteins of *C. nitidissima* and other plant species. The red and blue asterisks represent the active site and critical catalytic residues, respectively. The residues that affect substrate preference are indicated by the red box.

**Figure 4 biomolecules-13-00041-f004:**
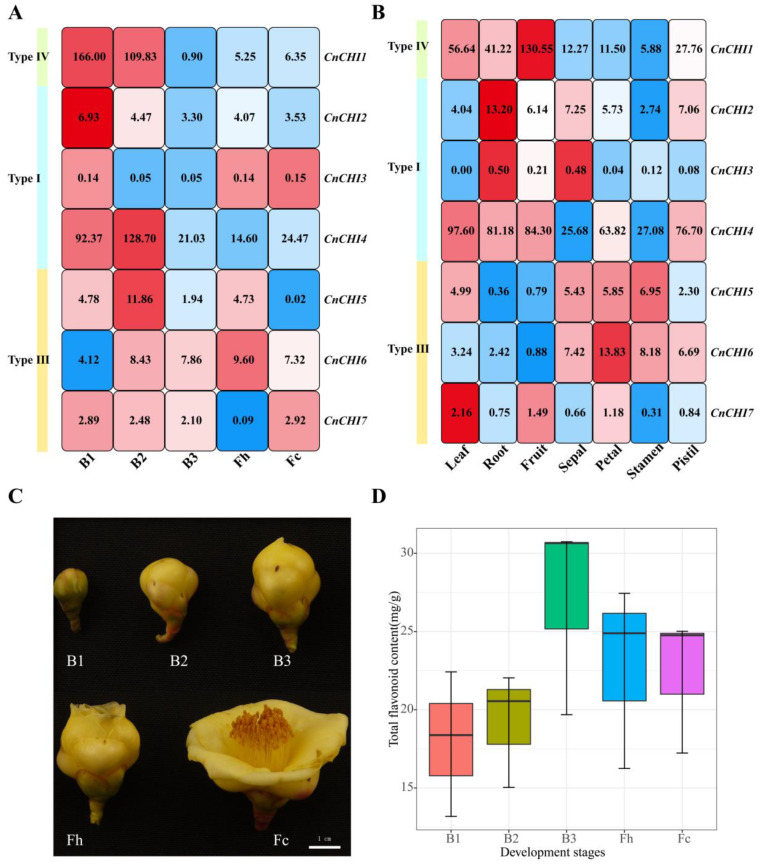
Expression profile of *CnCHI* and TFC in the petals of *C. nitidissima*. (**A**) Expression pattern of *CnCHI* in the petals of *C. nitidissima* at different developmental stages. B1, B2, and B3 represent buds 10 mm, 20 mm, and 30 mm in diameter, respectively; Fh, half-open flower; Fc, completely open flower. Numbers in the colored fields represent FPKM values. We normalized the data by standardizing each row and represented them by different colors, with blue representing low expression and red representing high expression (**B**) Expression pattern of *CnCHI* in the different tissues of *C. nitidissima*. (**C**) Petals of *C. nitidissima* at different stages of development. Bars = 1 cm. (**D**) TFC of the petals of *C. nitidissima* at different stages of development.

**Figure 5 biomolecules-13-00041-f005:**
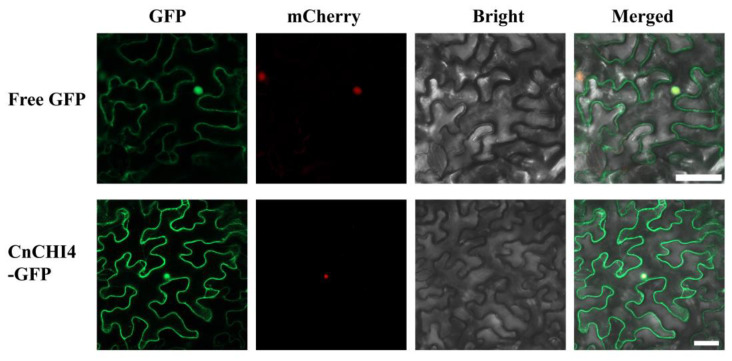
Subcellular localization of CnCHI4 in the leaves of *N. benthamiana*, co-expressed with nucleus-located mCherry. The empty vector was used as the control. GFP: images under green fluorescence channel; Bright: images under bright light; mCherry: images under red fluorescence channel; Merged: overlay plots. Bars = 50 μm.

**Figure 6 biomolecules-13-00041-f006:**
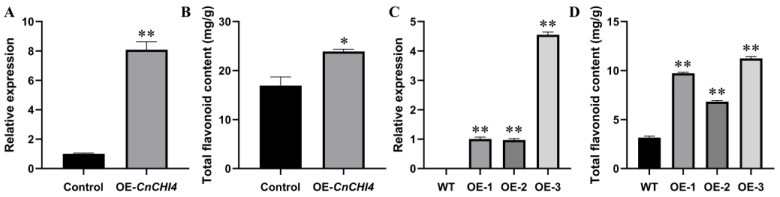
Overexpression of *CnCHI4* in *C. nitidissma* and *N. tabacum*. (**A**) Relative expression levels of *CnCHI4* in the control (empty vector) and experimental (pCAMBIA1302-*CnCHI4*) groups in *C. nitidissma* leaves after transiently transformed. (**B**) TFC in the control and experimental groups in *C. nitidissma* leaves after transiently transformed. (**C**) Relative expression levels of *CnCHI4* in the wild-type tobacco plants and plants overexpressing *CnCHI4*. (**D**) TFC in the wild-type tobacco plants and plants overexpressing *CnCHI4*. The error bars indicate the SE from three replicates; statistical significance was determined using Student’s t-test (* *p* < 0.05; ** *p* < 0.01). WT, wild-type, OE, overexpression.

**Figure 7 biomolecules-13-00041-f007:**
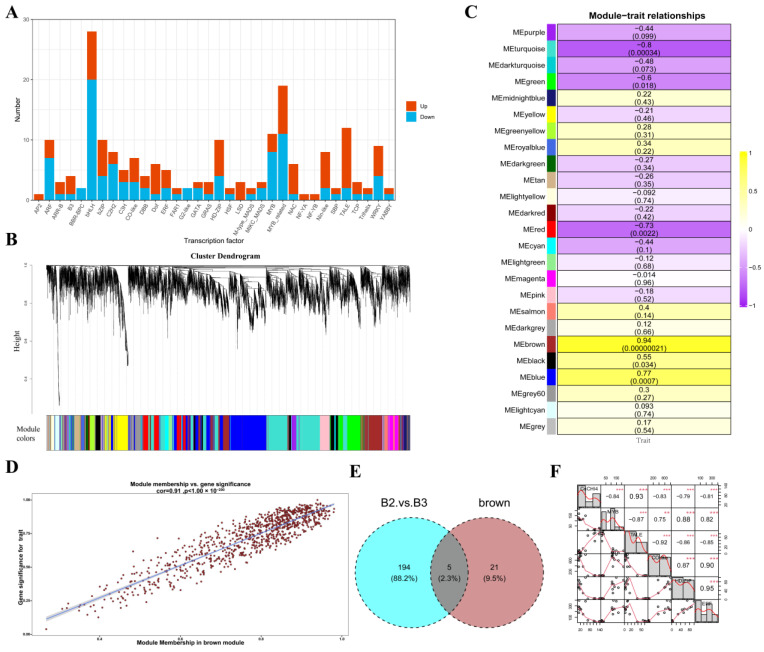
Identification of the transcription factors that regulate *CnCHI4* expression. (**A**) Transcription factors that were differentially expressed between B2 and B3; the red and blue bars represent upregulated and downregulated expression in B3 stage, respectively. ‘Transcription factor’ on the x-axis represents different transcription factor families, and ‘number’ on the y-axis means the number of DEGs detected within each gene family. (**B**) Hierarchical cluster tree of the co-expressed modules. (**C**) Correlation between the gene expression modules and the expression level of *CnCHI4* at different stages of flower development, the number in the brackets represents significance level. The bar ranging from purple to yellow (−1 to 1) represents negative to positive correlations, respectively. (**D**) Module Membership and Gene Significance of the genes in the brown module. (**E**) Venn diagram comparing the genes identified by WGCNA and differential expression analysis. (**F**) Pearson correlation coefficient between the expression of *CnCHI4* and key transcription factors at different stages of flower development. The statistical analysis was performed by R package, PerformanceAnalytics (** *p* < 0.01; *** *p* < 0.001).

**Figure 8 biomolecules-13-00041-f008:**
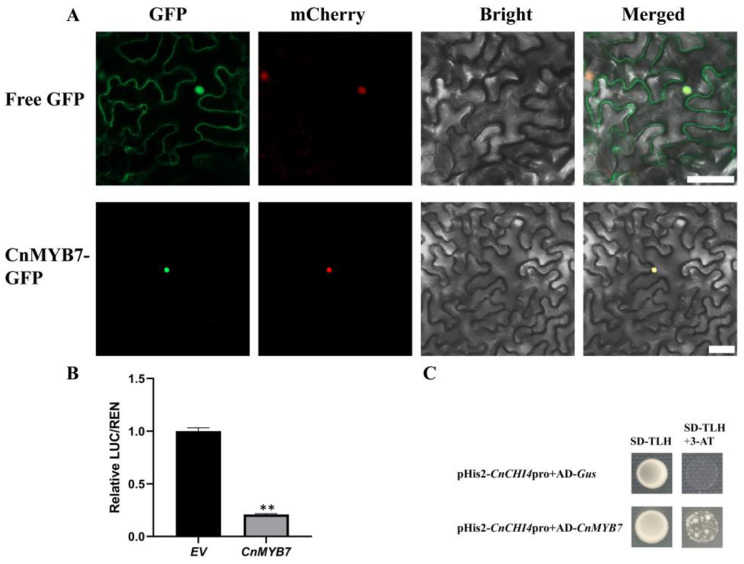
CnMYB7 regulates the expression of *CnCHI4.* (**A**) Subcellular localization of CnMYB7 protein in the leaves of *N. benthamiana*, co-expressed with nucleus-located mCherry. The empty vector was used as the control. GFP: images under green fluorescence channel; Bright: images under bright light; mCherry: images under red fluorescence channel; Merged: overlay plots. Bars = 50 μm. (**B**) Dual-luciferase reporter assay analyses the regulatory effects of CnMYB7 on the promoter of *CnCHI4*. The SE was determined from three replicates, and statistical significance was determined by Student’s *t*-test (** *p* < 0.01). (**C**) Yeast one-hybrid analyses of the interaction between CnMYB7 and the promoter of *CnCHI4*.

**Table 1 biomolecules-13-00041-t001:** Characteristics of CnCHI protein sequences.

Genes	Gene ID	Amino Acid Length (aa)	pI	Molecular Weight (KDa)	Grand Average of Hydropathicity	Instability Index	Signal Peptide
*CnCHI1*	F01.PB9454	209	4.83	23,193.41	−0.148	41.12	no
*CnCHI2*	F01.PB6499	219	5.39	23,711.19	−0.148	41.12	no
*CnCHI3*	F01.PB58445	225	5.25	24,264.77	−0.097	42.96	no
*CnCHI4*	F01.PB37720	230	5.25	24,735.25	−0.116	47.73	no
*CnCHI5*	F01.PB54606	293	8.73	32,154.61	−0.189	48.40	no
*CnCHI6*	F01.PB71691	431	8.6	47,966.94	−0.097	43.83	no
*CnCHI7*	F01.PB53164	452	8.6	50,117.33	−0.093	42.57	no

## Data Availability

The data presented in this study are available on request from the corresponding author.

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
