# Peer review of "Identification of Chalcone Isomerase Family Genes and Roles of CnCHI4 in Flavonoid Metabolism in Camellia nitidissima"

_biomolecules, 2022, doi:10.3390/biom13010041_

Round 1
Reviewer 1 Report
This manuscript analyzes chalcone isomerase family genes and especially the role of CHI4 in the flavonoid metabolism in Camellia nitidissima using protein analysis websites and experiments. The author uses a lot of experiments, however, the explanation is not enough for most of them.
1. " Singal peptide among 7 CnCHI proteins was also predicted using SignaIP-5.0, suggesting that the N-terminal signal peptide region was absent in these proteins." What is the difference between these signal peptides, the sequence difference, and the function difference or there is no difference? The author needs to give more explanation about this. Especially in figure 1, shows that almost all the CHIs have the difference conserved motifs at N-ter region except CHI1 and CHI5, then where is the signal peptide?
2. In figure 1, what are the conserved motif's name and function?
3. The letter in Fig. 2 is not clear and the Number in Fig. 3 is also not clear, please revise this.
4. In Fig.3, what is the sequence region of CnCHIs? From top sequence No. it is from about 260-480 aa, however, the amino acids of CHIs 1-7 do not have 480 aa. The author needs to explain how to make this alignment, and it is better if the author can combine this sequence alignment with the analysis of the conserved motifs.
5. In result 3.5, it is shown that CHI 4 locates in the cytoplasm and nucleus, which is different from WoLF PSORT prediction, why?
Reviewer 2 Report
Dear Authors,
I have reviewed your manuscript "Identification of chalcone isomerase family genes and the role of CnCHI4 in the flavonoid metabolism in Camellia nitidissima", submitted for publication in Biomolecules.
The results presented in your manuscript are novel and interested, and the research seems to be carried out properly. However, unfortunately, in its current version the manuscript is heavily flawed. I would advise you to take your time and thoroughly revise your manuscript to remove the numerous flaws, so that your interesting and important results can get a quality presentation which they deserve. Please consult my points for revision as given below:
· The transcriptome dataset – Your entire research relies on a transcriptome dataset that you briefly describe in lines 81-85, but, from what I can deduce from your paper, you have not published, or made available this transcriptome dataset in any way. Maybe you intended to publish your transcriptome dataset within another paper, which has not been published yet. Alternatively, you might have considered that publishing it was not important. I would urge you to wait with this paper until you publish your transcriptome research first, or, alternatively, publish your transcriptomic dataset within this paper.
· On a related note, two transcriptomic studies are already available where transcriptomes of flavonoid synthesis in Camellia nitidissima have been published: the paper by Zhou et al. (2017): https://doi.org/10.3389/fpls.2017.01545 and the paper by Liu et al. (2023): https://doi.org/10.1016/j.gene.2022.146924 Both of these papers are highly relevant to your study and deserve to be cited in the Introduction to your work. Additionally, you might decide to use their transcriptome datasets to verify that they match the results of your present research.
· the English language of your manuscript is heavily flawed, abunding with incorrect grammar, awkward formulations, and even poor spelling and typographic errors. The problem with the language is wider than the correct use of English itself; formulation such as "much work remaining from gene expression to metabolite synthesis" are unacceptable for a scientific publication. I urge you to employ a professional scientific language editing service to take care of your manuscript before resubmitting it to Biomolecules.
· several methods used in the research are not even hinted in the Material & Methods section (such as WoLF PSORT, SignalP 5.0, and so on). Please take care that all the methods that you employed in your research are mentioned and thoroughly described in your Material & Methods section. Please remember that the Material & Methods section is suppose to enable other researchers to replicate your research using the entirety of the same methods and bioinformatic tools that you used in your study.
· Certain results within your study are very poorly explained. I will give you some examples below:
o The results in Figure 4 (A and B) are very poorly explained. Is this relative expression? How is this relative expression calculated (relative to what)? What do the numbers in the colored fields mean? The color scales in both Figure 4A and 4B seem to be independent within every horizontal line (the numbers grow from dark blue to dark red only within a single horizontal line, but not between different horizontal lines within the same diagram) – however, this remains unexplained in the figure caption. What is the logarithmic base for the color scale in Figures 4A and 4B? Is it a log2, or a log10 scale? All these questions remain unanswered, making the results insufficiently clear to the reader if they are not thoroughly explained. On a separate note, the subfigures of Figure 4 (at least Figures 4A, 4B, and 4C) should be enlarged, because they are currently very poorly visible.
o The results of the WGCNA analysis in Figure 7 are also very poorly explained. You should remember that an important portion of your readers will not be familiar with how the WGCNA analysis is performed, and even for those who are, the results should be explained much more clearly. Currently, more than half of the explanation of how the results of the WGCNA were obtained and what they mean, is missing. Mentioning the "brown module" as if the meaning of "the brown module" was common knowledge, is just an example. Please remember, that the readers will not know what results you obtained in your study and how you are interpreting them, if you do not thoroughly explain it to them.
· A minor point: Please avoid the exaggerated statements, such as "Camellia nitidissima is one of the rarest plants in the world" (beginning of Abstract; it is definitely not one of the rarest plants in the world because if it was, we likely wouldn't even be aware of its existence) or "flavonoids are extremely beneficial in the promotion of health" (line 395; flavonoids are definitely beneficial for health, but it would be hard to prove that they are "extremely beneficial", because there is no good criterium for "extremely beneficial").
I hope that my remarks will not discourage you, but instead help you re-write your manuscript and present the results of your research in a more appropriate way.
Kind regards,
Reviewer
Round 2
Reviewer 1 Report
The authors addressed my concerns.
Author Response
Thank you very much for your work.
Reviewer 2 Report
Dear Authors,
I have been asked for the second round of review of your manuscript "Identification of chalcone isomerase family genes and roles of CnCHI4 in flavonoid metabolism in Camellia nitidissima", submitted for publication in Biomolecules.
Your revised manuscript presents considerable improvements compared to the initial submission, including a proper reference to the full transcriptome dataset and a mostly adequate revision of English language. Moreover, some of the methods that were previously missing from the Methods section have now been added, and the results in Figures 4 and 7 are now somewhat better explained than in the initial submission.
Nevertheless, despite considerable improvement, there still remains work to do before the manuscript can be considered for publication. The Methods section needs further corrections and additions, including the cultivation conditions for the C. nitidissima plants, the methodology for the absolute quantification of gene expression, and the quantification of total flavonoid content. The results concerning gene expression shown in Figure 4 are still inadequately presented, and besides, an explanation of the quantification of absolute gene expression is missing from both the Figure caption, and the Material & Methods section. The results shown in Figure 7 have been somewhat improved, but more unconnected dots remain to be explained. Besides, although the citations of the two works already published on the transcriptomics of flavonoid synthesis in C. nitidissima (Zhou et al. 2017; Liu et al. 2023) have been added to the Introduction, the important information from these works, that is relevant to the aim of your present work, has not been communicated to the readers.
When submitting your revised manuscript, I would ask you to please remove the deleted parts of the text. They act distracting and make it difficult to smoothly read the text. For particular corrections, please consult my points as given below:
· Figure 4 – The explanations for Figure 4 have been somewhat improved, but not sufficiently. It is good that you stated that the color plots are applied to each horizontal row independently of each other; however, you did not address most of my other questions related to that Figure. Please add the following explanations to the Figure captions:
o What is the logarithmic base for the color scale in Figures 4A and 4B? Is it a log2, or a log10 scale? Please provide an explanation within the figure caption.
o In your response to me you stated that you improved the quality of the picture for Figure 4. However, I haven't noticed that the subfigures were increased in size. The lettering is still very small, and also the pictures of the flowers in 4C are tiny, yet there is still enough space around the pictures to increase their size. Please take care of resizing the subfigures.
o FPKM (fragments per kilobase million) needs to be fully spelled out in line 376.
o The calculation of the FPKM values, including the full quantification method and the standards that you used for the quantification calibration curves, need to be fully explained in the Material & Methods section.
o Within the manuscript text, the results presented in Figure 4B (lines 387-412) are introduced before the results presented in 4A (lines 412-436). This makes no sense. Please switch the order, either of subfigures A and B, or the order in which you introduce them within the manuscript text.
· Figure 7 – Although the WGCNA analysis is now somewhat better explained in the text, parts of explanation are still missing, especially in the figure caption. Also, all the subfigures should be appropriately cited in the text of the Results, to facilitate the smooth following of the information from the graphs to the reader:
o line 500: please add the citation of Figure 7C at the end of the sentence "(r = 0.94, p = 2.1 x 107)"
o line 502-504: Please bring back the part of the text where the meanings of the abbreviations GS and MM are explained.
o line 504: please add the citation of Figure 7D at the end of the sentence, after "MM".
o line 517: please add "either positively or negatively" ("were significantly associated, either positively or negatively, with the expression of CnCHI4")
o Figure caption for 7A: I guess that "number" on the vertical axis means the number of DEGs detected within each gene family? Please spell out the explanation within the figure caption.
o Again, figure caption for 7A: What does "upregulated" and "downregulated" mean? Upregulated or downregulated in B2 compared to B3, or in B3 compared to B2? The way it is currently written, it could be interpreted either way. Please spell out the explanation within the figure caption. Alternatively, you may change the explanation of the red and blue boxes with "upregulated in B2" and "upregulated in B3".
o Figure caption for 7C: Two numbers are given within each colored box. The first number is clearly the Pearson's correlation coefficient. But what is the number in the brackets? Please explain it within the figure caption.
o Again, figure caption for 7C: What is the logarithmic base for the yellow-purple color scale? Is it a log2, or a log10 scale? Please provide an explanation within the figure caption.
o Figure caption for 7D: Please fully spell out GS and MM even if you also spell them out in the manuscript text.
· Material & Methods – important parts of methodology for your research are still missing from your manuscript. Please take care of the following:
o line 196-197: total flavonoid content cannot be the measure of quantification of gene expression, no matter which gene it is. The measurement of total flavonoid content needs to be fully described, in a separate (new) subsection of Material & Methods. The complete method needs to be described, including how the material was sampled, from what tissues and at what age, which methodology was used for extraction, and so on.
o section 2.8: the calculation of the FPKM values, including the full quantification method and the standards that you used for the quantification calibration curves, need to be fully explained in the Material & Methods section.
o line 231: A serious part of methodology regarding plant growth conditions is missing here. The leaves were obtained from the National Germplasm Resource Bank? The leaves, or the whole plants from which the leaves were sampled? How were the plants cultivated? In vitro, or in soil pots in a greenhouse? Under what temperature, humidity, and light regime? How were the leaves sampled?
· Background information and aim of your work – It is good that you have introduced the citations of the two previous works on the transcriptomics of flavonoid biosynthesis in C. nitidissima (references 29 and 30). However, the findings present in these papers are not yet adequately reflected within the text of your manuscript (both Introduction, and Discussion). This is very important, since your current work needs to be put in the context of what is already established about the flavonoid biosynthesis in your research object. I will give you some very particular advice on how to address that:
o Introduction:
§ line 90-92: Please delete this part: "but there is scarcity of information regarding the regulatory effect of transcription factors on flavonoid synthesis in Camellia nitidissima"
§ line 99-100: Please replace the sentence: "However, there is scarcity....... in C. nitidissima" with a more appropriate statement, where you would (very briefly) introduce the findings from the works of Zhou et al. 2017 [29] and Liu et al. 2023 [30]. In both papers, transcriptomic research of flavonoid biosynthesis in C. nitidissima was performed. In the first paper [29], no important involvement of CHI genes was found in the flavonoid biosynthesis in C. nitidissima, whereas in the second study [30] it was found that the CHI genes were involved in flavonoid biosynthesis during the early stages of flower development. This coincides with the findings of your research and therefore needs to be stated in the Introduction (and later, in the Discussion). Then, you may frame the aim of your present research as wanting to more closely examine the roles of the CHI genes in flavonoid biosynthesis during the flower development of this species. The transcriptomic research cited [30] was not specifically oriented towards the function of CHI genes, so you want to address this topic more in detail – as you already correctly stated in line 102. This is generally how the aim of your research should be framed.
o Discussion:
§ line 571: The context provided by the previous works [29,30] should be reiterated here. I suggest a wording more or less like this: "Previous transcriptomic studies did not reveal major specific roles for CHI genes in the flavonoid metabolism of C. nitidissima [29,30], although recently it was shown that these genes are upregulated in the petals during the early stages of flower development [30]."
§ line 631-632: Please add: "and in a recent study involving C. nitidissima [30]". This is much more important for your paper than Antirrhinum majus or citruses. Also please put back the normal full spelling of Antirrhinum majus.
· Other remarks:
o lines 61, 64: there is a L missing in "hydroxyflavone", please revise
o line 96: please correct "components" to "compounds"
o line 98: please put "genes" and "flavonoids" into plural
o line 158-159: the weblink for SOPMA does not seem to work, please double-check whether the web address is right
o lines 200: please correct "Ag. sp." to "A. tumefaciens"
o line 212: please bring back the deleted part "genes (DEGs)" into the subsection title
o line 236: Cj, or Cn?
o line 262: Strain Y187, please state which species of yeast?
o line 293: From the Table, one would say that the pI values ranged from 4.83 for CnCHI1, to 8.73 for CnCHI5; not from 4.76 to 8.6. Please double-check and revise.
o lines 304-307: Is a visualization of the secondary structure predictions for the 7 CnCHI proteins available? It would make for a good addition as a new figure.
o line 387: please correct: "ARE displayed in a heatmap"
o line 391: please correct: "do not play AN IMPORTANT role" (you cannot know that they don't play a role at all)
o line 439: The total flavonoid content, in which tissue? This needs to be spelled out here in the Results section, even if it has already been said in Material & Methods.
o line 447: "appeared to be somewhat similar to the expression" – please correct to: "appeared to reflect the expression"
o line 453-454: this part of the text is unnecessarily long. Please replace it with: "The findings demonstrated that CnCHI4 is located in the cytoplasm and nucleus."
o line 479 and line 561: CnCHI4 should be in italic letters.
o line 502-504: Please bring back the part of the text where the meanings of the abbreviations GS and MM are explained.
o line 567: Please add the word "flowers" ("C. nitidissima flowers")
o line 584-585: Please revise into: "which was also found in leguminous plants". The way it is worded right now, makes it sound as if Camellia was itself a leguminous plant as well.
o line 594: Please correct "and contain Ser and Ile" into "including Ser and Ile"
o line 624, 650, 659: Please correct "Ar. sp." into "A. thaliana"
o line 631-632: Please add: "and in a recent study involving C. nitidissima [30]". This is much more important for your paper than Antirrhinum majus or citruses. Also please put back the normal full spelling of Antirrhinum majus.
o line 642: Please bring back the expression pattern instead of alterations: "These findings explain the expression pattern..."
o Author Contribution Statement: please write the statement in line with the CRediT Taxonomy guidelines: https://credit.niso.org/
